# The Unique Experience of a New Multidisciplinary Program for 22q Deletion and Duplication Syndromes in a Community Hospital in Florida: A Reaffirmation That Multidisciplinary Care Is Essential for Best Outcomes in These Patients

**DOI:** 10.3390/genes13111949

**Published:** 2022-10-26

**Authors:** Zaimary Meneses, Jenna Durant, Hanadys Ale

**Affiliations:** 1Charles E. Schmidt College of Medicine, Florida Atlantic University, Boca Raton, FL 33431, USA; 2Division of Immunology, Allergy, and Rheumatology, Joe DiMaggio Children’s Hospital, Memorial Healthcare System, Hollywood, FL 33021, USA

**Keywords:** 22q11.2 deletion syndrome, 22q11.2 duplication syndrome, multidisciplinary clinic, multidisciplinary care, chromosomal microdeletion

## Abstract

In 2018, the first 22q11.2 multidisciplinary program in the state of Florida was created at Joe DiMaggio Children’s Hospital following the new paradigm for best care of 22q11.2 deletion patients. Since inauguration, the clinic flourished despite challenges. Our 22q clinic has 149 patients ranging from ages 0–21. From that total, 138 are 22q11.2DS: 74 females and 64 males (44% Hispanics, 35% Caucasians, 11% African American, 3% Asian and 7% multiracial). Eleven patients are in the 22q11.2 duplication group; 7 females and 4 males (50% Hispanics, 30% Caucasians 10% Asian and 10% multiracial). Our multidisciplinary team has grown to include twelve different specialties to better serve our growing patient population and has adapted to the pandemic by offering virtual clinics. Although there are many 22q multidisciplinary clinics worldwide, our clinic has special characteristics. We have an ethnically diverse group of patients and a large team of mostly bilingual providers who are passionate about and have expertise on 22q Deletion/Duplication Syndromes. Our 22q clinic is based at a community hospital and counts on the partnership of local 22q patient support groups. The program is also unique in that it is now expanding to care for adult 22q patients. Our clinic is another live example of how multidisciplinary care is the best way to achieve the most optimal outcomes in 22q patients, and that if there is a passionate and dedicated team of providers willing to collaborate for these patients, a 22q multidisciplinary program can thrive, succeed and grow at a community hospital.

## 1. Introduction

22q11.2 Deletion Syndrome (22q11.2DS) is the most frequent chromosomal microdeletion syndrome and is the second most common cause of developmental delay and congenital heart disease, following Down syndrome [1]. 22q11.2 has become an umbrella term to describe multiple other syndromes that were previously described separately but shared similar characteristics, such as DiGeorge Syndrome, Velocardiofacial Syndrome, and Conotruncal Anomaly Face Syndrome [2,3]. With the advancement and increased accessibility to genetic testing and recognition of specific chromosomal abnormalities, we have been able to better characterize the condition as one collective syndrome with varying presentations: 22q11.2 deletion syndrome [3,4].

The prevalence of 22q11.2DS is estimated to be within 1 per 2148 livebirths and 1 in 992 unselected pregnancies, however the actual incidence may be greater given the variable expressivity of the microdeletion [2,5,6]. Based on our calculations, in the state of Florida there should be about 10,000 patients with 22q11.2DS based on a population of 21.78 million people in the state, with 40% of them residing in South Florida [7]. Additionally, this does not account for patients with 22q11.2 duplication syndrome. This highlights the importance of a multidisciplinary 22q clinic in the South Florida region.

22q11.2DS is highly variable and has a wide range of phenotypic manifestations. Multiple organ systems are affected in these patients, with no two individuals having the exact same phenotypic presentation. Patients can present clinically with cardiac abnormalities, palatal abnormalities, immune deficiencies, endocrine abnormalities, genitourinary and gastrointestinal involvement, developmental delay, cognitive deficits, and psychiatric conditions [2]. Given the complexity of each individual patient, the unique needs of each one of them, and the spectrum of specialties who manage the care of patients with 22q11.2DS, there has been a change in paradigm and a multidisciplinary approach is becoming the standard of care to optimize outcomes and accessibility of care for these patients and their caregivers [5].

Similarly, 22q11.2 duplication syndrome, the reciprocal syndrome of 22q11.2DS also demonstrates phenotypic variability ranging from asymptomatic to dysmorphic features, cardiac malformations, and neurodevelopmental delays [7]. Although 22q11.2 duplication syndrome is less prevalent than its counterpart 22q11.2DS, the use of microarray technology has led to increased rates of identification [8].

Before the existence of 22q multidisciplinary clinics, patients with 22q11.2 deletion and duplication syndromes received fragmented and incomplete care. However, after the first multidisciplinary clinic was created, it was clearly noted that this model would benefit patients as well as their caregivers and physicians. Moreover, being part of an interdisciplinary network facilitates providing timely intervention, monitoring, and support to patients, as well as physician access to a network of peers [5]. Moreover, there is a set of practice guidelines which outlines the standard of care for 22q11.2DS patients and a timeline for screening and monitoring, which is implemented and followed at each multidisciplinary clinic.

## 2. Materials and Methods

In 2018, a team at a community pediatric hospital in South Florida, Joe DiMaggio Children’s Hospital, created the first 22q11.2 multidisciplinary clinic in the state of Florida. The original director of this program trained at Children’s Hospital of Philadelphia (CHOP), the home of the 22q and You Center, and had a vision of creating a multidisciplinary team in her new institution in Florida. She implemented a similar model as CHOP to best care for the number of 22q patients seen at Joe DiMaggio. The inaugural team consisted of genetics/genetic counseling, immunology, ENT, endocrinology, speech pathology, and nutrition. All inaugural members had previous experience with 22q patients and a full understanding of their medical needs. Amidst a change in leadership and the COVID-19 pandemic, the clinic faced hardships and was close to being dissolved. However, the team persevered through the challenges and came out stronger.

The 22q Program has grown and flourished and is now a total of 14 providers with the addition of six other specialties: cardiology, neurology, psychology, nephrology, social work, and pulmonology. We analyzed the specific breakdown of involvement of each organ system and each specialty’s involvement in our patients’ care via a database containing information from our cohort of patients. Our main patient referral sources are genetics, cardiology, ENT, and speech pathology. The 22q clinic is held every third Wednesday of each month and the core multidisciplinary team (genetics, endocrinology, ENT, speech pathology, immunology, and social work) sees a maximum of six patients per clinic day. A post-clinic conference is held right after each clinic to discuss the specialty recommendations for all patients seen that day, as well as the appropriate referrals. The patient census has also grown exponentially in the past four years.

## 3. Results

Our 22q clinic has evaluated 149 patients ranging from 0–21 years of age. From that total, 138 have 22q11.2DS: 74 females and 64 males (44% Hispanic, 35% Caucasian, 11% African American, 3% Asian and 7% multiracial). Eleven patients are in the 22q11.2 duplication group; 7 females and 4 males (50% Hispanic, 30% Caucasian, 10% Asian and 10% multiracial). In our patient population, the breakdown of organ-system-specific symptomatology is as follows: 72% cardiac, 60% endocrinologic, 58% ENT, 62% immunologic, 64% speech, 59% neurologic, 47% pulmonological, 44% nephrological. The percentages from specialties which we still need to incorporate into the clinic, however with which patients have reported issues, are as follows: 59% psychiatric, 46% gastrointestinal, 40% dental, and 37% orthopedic. The phenotypic presentation of our patient population is very heterogeneous and spans almost all described clinical manifestations, although we have not had any patients with complete DiGeorge Anomaly (athymia, congenital heart disease, and hypoparathyroidism). Equally as diverse as the phenotypic characteristics of our patients are their genotypic presentations. Most patients have typical 2.5 Mb deletions (between LCRs A-D) and a minority of patients have 1.5 Mb deletions (LCRs A-B) and atypical deletions within LCRs A-D. However, we do not have patients with point mutations in the TBX1 gene. We also care for two sisters with 22q11.2 duplications, one with a 6.13 Mb interstitial duplication and one with a 6.4 Mb interstitial duplication, which to our knowledge have not been previously described in the literature.

Our clinic has adopted initiatives to better serve our unique population and to expand our program. To initiate care as early as possible, our multidisciplinary program is collaborating with the neonatology/maternal-fetal medicine specialties to capture the prenatal diagnosis of 22q and participate in counseling the families. We are also establishing partnerships with our adult counterparts with the plans to mirror our pediatric multidisciplinary clinic. We are hoping to launch our new adult version of the 22q clinic in January 2023 with the following specialties: cardiology, genetics, endocrinology, immunology, and social work. This way we can better serve the patients already diagnosed by our adult colleagues, our own patients reaching adulthood, and those parents that we capture via parental genetic testing. The 22q multidisciplinary clinic has also adapted to the challenges of COVID-19 by offering virtual clinics via Webex.

## 4. Discussion

Multidisciplinary care for patients with 22q11DS has been described in the literature and has been shown to have positive outcomes. For example, in a Quality Improvement project by Hickey et al., patients cared for at a multidisciplinary 22q Center had a guideline adherence rate of 83.0% compared to 41.7% of patients not receiving care at a multidisciplinary clinic [1]. Aside from the positive outcomes of multidisciplinary care for patients with 22q11DS, there have been similar studies showing positive outcomes for patients with other genetic syndromes, such as Down syndrome. One study by Skotko et al. showed that providing care through a multidisciplinary clinic increased the likelihood of patients with Down Syndrome receiving care as per recommended guidelines [9]. Thus, multidisciplinary care has proven itself to be superior to traditional care for various genetic conditions.

The need for multidisciplinary care for patients with complex medical conditions has been well-documented in the literature for several conditions including Down syndrome, multisystem inflammatory disease in children (MIS-C), X-linked hypophosphatemic rickets, Marfan syndrome, Cystic Fibrosis, among many others [10,11,12,13]. The complexity and expressivity of 22q11.2DS and 22q11.2 duplication syndromes are no exception to the need for multidisciplinary care. Across the globe, multiple studies have highlighted the need for interdisciplinary clinics for patients with 22q syndromes, in addition to the successes of the established clinics [14,15,16,17].

Although there are many 22q multidisciplinary clinics worldwide, our clinic has unique characteristics. Being in the South Florida area, we have a very ethnically diverse group of patients and a large team of mostly bilingual providers who are passionate about and have extensive expertise on 22q deletion and duplication syndromes. Our 22q clinic is also unique in that it is based out of a community children’s hospital. The clinic relies on local 22q patient support groups and has formed partnerships with local 22q family organizations. Our 22q Program is continuously working with various local support centers to raise awareness, like the Florida International University Embrace Center. This is a center with 22q members that empowers differently abled people to live their fullest potential via research, awareness in the community, and developing their professional skills. Through partnerships such as this one, we are receptive to their feedback and are continuously striving to improve our clinic. Similarly, as part of our tactic to raise awareness about 22q11.2 deletion and duplication syndromes, the clinic has presented at local medical schools, planned townhalls with community pediatricians, and is planning to execute outreach projects to the Caribbean and Latin America where there is a significant lack of 22q multidisciplinary clinics.

We feel that our 22q multidisciplinary program has improved the quality of life of our patients and their caregivers and has assisted in delivering accessible, high-quality care. By making the clinic our patients’ “medical home” we have offered them a space to feel taken care of and understood. Similarly, we strongly believe in the importance of acting early, becoming involved in our patients’ care as soon as the diagnosis is made, and always thinking ahead in order to avoid adverse outcomes. We want to emphasize the importance of individualized care, as no two 22q patients display the exact same phenotypic manifestations, even amongst relatives. As shown by the breakdown of organ system-specific manifestations in our results, the care provided to our patient cohort is very heterogeneous. While we currently have involvement from cardiology, endocrinology, ENT, immunology, speech pathology, neurology, pulmonology, and nephrology, we also want to highlight the importance of further collaboration from other specialties. As shown in our results, our patients have also reported issues involving psychiatry, GI, dental, and orthopedics. It is our goal at our 22q multidisciplinary clinic to deliver seamless care from various physicians and specialists in order to deliver anticipatory guidance and optimize outcomes for our patients. Through our patients and their caregivers, we know that having a one-stop destination for their medical care has improved their quality of life and well-being.

Additionally, our children’s hospital is neighbored by its adult counterpart, Memorial Regional Hospital. Being that our healthcare system also has an adult patient population, the multidisciplinary team has now undertaken the mission to include adult patients within the 22q multidisciplinary clinic in the future. Since pediatric patients transition into the adult world between the ages of 18–21 years old, we feel that it is important to continue caring for adult 22q patients within the interdisciplinary setting in order to provide longitudinal care. Adults with 22q syndromes continue to have unique medical needs and may have complex conditions spanning across various organ systems, such as psychiatry, gynecology, and nutrition, and thus an interdisciplinary team can have benefits which translate into the adult realm as well [18].

## 5. Conclusions

Being that our clinic has thrived within our community hospital setting and is continuously improving, we want to emphasize that multidisciplinary clinics should be the standard of care for patients with 22q11 deletion and duplication syndromes, regardless of setting. We feel that given the proper catalysts, a 22q multidisciplinary clinic is possible and can significantly improve outcomes for the patients they serve. These factors include a highly motivated and dedicated team with expertise in the field, a setting with access to various subspecialties, and a location within a sizeable patient population, a very likely occurrence given the prevalence of the syndrome. The presence of 22q multidisciplinary programs can also act as a link to promote connection and support amongst families with children with 22q syndromes through hosting community events, such the local 22q at the Zoo.

To continue to provide individualized and holistic care for our patients some additional studies may be helpful in highlighting the need for further specialty incorporations like orthopedics, dental, and gastrointestinal. A quality improvement study can be done via patient and family surveys and questionnaires inquiring about how to improve the experience of being part of the multidisciplinary clinic. Furthermore, collaboration amongst the established 22q programs should be encouraged, as well as providing support for emerging 22q clinics with the expertise of the existing ones.

## Data Availability

Not applicable.

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
