# Peer review of "The Unique Experience of a New Multidisciplinary Program for 22q Deletion and Duplication Syndromes in a Community Hospital in Florida: A Reaffirmation That Multidisciplinary Care Is Essential for Best Outcomes in These Patients"

_genes, 2022, doi:10.3390/genes13111949_

Round 1

Reviewer 1 Report

Article titled "The Unique Experience of a New Multidisciplinary Program for 22q Deletion and Duplication Syndromes in a 2 Community Hospital in Florida: A Reaffirmation that Multidisciplinary Care is Essential for Best Outcomes in 3 these Patients" is interesting and highlights the need for a multidisciplinary approach to a condition characterized by pleiotropy and variable expressivity.

I suggest that it would be interesting to add the degree of involvement of various specialties (percentages of patients with cardiac symptomatology in your group for example) in the management of the mentioned syndromes. In this way, the role of each specialty in the multidisciplinary team would be emphasized once again.

Author Response

As per review report 1: We have updated our “methods” to describe how we attained a breakdown of our patient cohort’s symptomatology and specialty involvement. We have now included this percentage breakdown in our results section. We have also updated our discussion to reflect the updated information.

Reviewer 2 Report

The topic is not innovative since the need for multidisciplinary care for 22q11.2DS patients is a well known concept. However, I find the idea of sharing the organizational model of Joe DiMaggio Children’s Hospital, and the results obtained from it, very useful. The paper is well written and I have only a few suggestions.

Introduction: the section ranging from line 73 to line 82 should be moved to the discussion

I suggest strongly emphasizing two aspects:

The pediatric immunologist should always be part of the multidisciplinary team

The need for adult patients multidisciplinary team, possibly in the same hospital. In particular, the psychiatrist should be part of the equipe for adult patients, together with the gynecologist and the nutritionist. Some clinical issues of patients in transitional age should be mentioned (J Allergy Clin Immunol. 2020 Nov;146(5):967-983).

Author Response

As per review report 2: We have moved lines 73-83 from the introduction section to the first paragraph of the discussion section. We have emphasized the need for the pediatric immunologist. We have also updated the specialties important for involvement in the adult multidisciplinary clinic.